# T Cell Homeostasis Disturbances in a Cohort of Long-Term Elite Controllers of HIV Infection

**DOI:** 10.3390/ijms25115937

**Published:** 2024-05-29

**Authors:** José M. Benito, Daniel Jiménez-Carretero, Clara Restrepo, José M. Ligos, Jaime Valentín-Quiroga, Ignacio Mahillo, Alfonso Cabello, Eduardo López-Collazo, Fátima Sánchez-Cabo, Miguel Górgolas, Vicente Estrada, Norma Rallón

**Affiliations:** 1HIV and Viral Hepatitis Research Laboratory, Instituto de Investigación Sanitaria Fundación Jiménez Díaz, Universidad Autónoma de Madrid (IIS-FJD, UAM), 28040 Madrid, Spain; clara.restrepo@hospitalreyjuancarlos.es (C.R.); norma.rallon@fjd.es (N.R.); 2Hospital Universitario Rey Juan Carlos, 28933 Móstoles, Spain; 3Unidad de Bioinformática, Centro Nacional de Investigaciones Cardiovasculares (CNIC), 28029 Madrid, Spain; daniel.jimenez@cnic.es (D.J.-C.); fscabo@cnic.es (F.S.-C.); 4Cytek Biosciences, Inc., Fremont, CA 94538, USA; jligos@cytekbio.com; 5Grupo de Respuesta Inmune Innata, IdiPAZ, Hospital Universitario La Paz, 28046 Madrid, Spain; jaimevquiroga@hotmail.com (J.V.-Q.); elopez.hulp@salud.madrid.org (E.L.-C.); 6Department of Statistics, Instituto de Investigación Sanitaria Fundación Jiménez Díaz, Universidad Autónoma de Madrid (IIS-FJD, UAM), 28040 Madrid, Spain; imahillo@fjd.es; 7Hospital Universitario Fundación Jiménez Díaz, 28040 Madrid, Spain; acabello@fjd.es (A.C.); mgorgolas@fjd.es (M.G.); 8Hospital Universitario Clínico San Carlos, 28040 Madrid, Spain; vicente.estrada@salud.madrid.org

**Keywords:** HIV control, long-term elite controllers (LTECs), T cell homeostasis, immunology

## Abstract

Elite controllers (ECs) are people living with HIV (PLWH) able to control HIV replication without antiretroviral therapy and have been proposed as a model of a functional HIV cure. Much evidence suggests that this spontaneous control of HIV has a cost in terms of T cell homeostasis alterations. We performed a deep phenotypic study to obtain insight into T cell homeostasis disturbances in ECs maintaining long-term virologic and immunologic control of HIV (long-term elite controllers; LTECs). Forty-seven PLWH were included: 22 LTECs, 15 non-controllers under successful antiretroviral therapy (onART), and 10 non-controllers not receiving ART (offART). Twenty uninfected participants (UCs) were included as a reference. T cell homeostasis was analyzed by spectral flow cytometry and data were analyzed using dimensionality reduction and clustering using R software v3.3.2. Dimensionality reduction and clustering yielded 57 and 54 different CD4 and CD8 T cell clusters, respectively. The offART group showed the highest perturbation of T cell homeostasis, with 18 CD4 clusters and 15 CD8 clusters significantly different from those of UCs. Most of these alterations were reverted in the onART group. Interestingly, LTECs presented several disturbances of T cell homeostasis with 15 CD4 clusters and 13 CD8 clusters different from UC. Moreover, there was a specific profile of T cell homeostasis alterations associated with LTECs, characterized by increases in clusters of naïve T cells, increases in clusters of non-senescent effector CD8 cells, and increases in clusters of central memory CD4 cells. These results demonstrate that, compared to ART-mediated control of HIV, the spontaneous control of HIV is associated with several disturbances in CD4 and CD8 T cell homeostasis. These alterations could be related to the existence of a potent and efficient virus-specific T cell response, and to the ability to halt disease progression by maintaining an adequate pool of CD4 T cells.

## 1. Introduction

HIV infection represents a paradigm of chronic viral infection without a model of spontaneous clearance. A subgroup of people living with HIV (PLWH) known as elite controllers (ECs) [1] are the closest model of this situation; they have the ability to limit viral replication to undetectable levels in the absence of treatment. These PLWH are thought to be the best example of a functional cure for HIV infection because of ECs’ remarkable capacity to control HIV replication and disease progression. In this scenario, the virus is greatly suppressed but not completely eradicated, allowing the patient to maintain immunological and virological stability without needing to take life-long antiretroviral therapy (ART) [1,2].

Since ECs have this ability, numerous studies have been conducted to identify the immunological mechanisms underlying the spontaneous control of HIV disease [3]. This is necessary to identify surrogate immunological markers of protection against the progression of HIV disease, which is a necessary first step in developing immunotherapeutic strategies aimed at controlling the infection without the need for life-long ART. The immune response mediated by T cells, particularly CD8 T cells, has been one of the most extensively researched of the proposed mechanisms because of its significant function in controlling viral replication [4,5,6,7,8].

However, the fact that ECs are a diverse population of PLWH [9] has complicated efforts to identify correlates of HIV disease control using ECs as a model. A significant proportion of ECs are unable to sustain HIV control over extended periods of time, and a number of studies have found that EC cohorts experience varying rates of virological and/or immunological control loss [10,11]. Several factors have been associated with the loss of virological and/or immunological control in ECs, including a waning HIV-specific T cell response [12], a heightened inflammatory milieu [13], and alterations in several aspects of T cell homeostasis [14]. In fact, the percentage of ECs able to maintain long-term immunological and virological control—known as long-term elite controllers, or LTECs—may be as low as 0.15% of the entire PLWH population [11]. Therefore, research aiming at identifying the immunological traits associated with protection against HIV disease should concentrate on the population of LTECs, as they represent the closest model of an HIV functional cure.

Given the pivotal role of T cells in maintaining not only an adequate anti-viral immune response but also the homeostatic equilibrium necessary to halt immunological progression, the goal of the current study was to thoroughly characterize the T cell phenotypes in a well-defined cohort of LTEC patients in comparison to both non-controller patients whose viral replication was uncontrolled and non-controller patients whose viral replication was controlled by ART.

## 2. Results

### 2.1. Characteristics of the Study Groups

A total of 67 volunteers were included in the study, including 47 PLWH volunteers and 20 HIV-seronegative volunteers (UC group). Among the PLWH, 22 were LTECs, 15 were onART, and 10 were offART. Table 1 shows the main characteristics of the different study groups at inclusion. Age was similar in all study groups, whereas the distribution of sex was different, with a majority of males in the onART and offART groups, in contrast to a more balanced distribution of males and females in the LTEC and UC groups. CD4 counts were similar among the different PLWH groups. Length of HIV infection (since diagnosis) was lowest for the offART group. The length of EC status in the LTEC group was 13 years and the length of ART in the onART group was 6 years.

### 2.2. Unsupervised Multidimensional Analysis of Flow Cytometry Data Yielded Several CD4 and CD8 T Cell Clusters That Differentiated the Study Groups

Spectral flow cytometry was used to examine eighteen CD4 and CD8 T cell markers in order to investigate various functional traits of these cells: maturation stage (CD45RA, CCR7, and CD28 markers); recent thymic emigrants (RTEs) (CD31 marker); activation (CD38 and HLADR markers); exhaustion (PD-1, Tim3, CTLA4, and TIGIT markers); senescence (CD57 and CD28 markers); apoptosis (CD95 and CD28 markers); homeostatic potential (CD127 marker); Tregulatory (Treg) cells (CD127, CD25, and CD39 markers); Thelper or Tcytotoxic 1 (Th1 or Tc1), Thelper or Tcytotoxic 2 (Th2 or Tc2), and Thelper or Tcytotoxic 17 (Th17 or Tc17) cells (CXCR3 and CCR6 markers); and peripheral follicular helper T cells (pTfh) (CXCR5 marker) (Appendix A). To create a two-dimensional map and reduce the dimensionality of the data, T-distributed stochastic neighbor embedding (tSNE) was used. The normalized density of events across this tSNE map was different in the four study groups, as well as the distribution of events from each group in the tSNE map (Appendix A), suggesting the existence of differences in the phenotypes of CD4 and CD8 T cells between the study groups. Clustering analysis yielded 57 clusters of CD4 and 54 clusters of CD8 T cells (Appendix A), differentiated by the expression levels of the markers analyzed (Appendix A). The proportion of each cluster in the total population of CD4 or CD8 T cells was calculated for each study sample and a differential abundance analysis was performed, using the UC group as a reference.

Several clusters of CD4 (Figure 1 and Appendix A) and CD8 (Figure 2 and Appendix A) T cells were differentially expressed in the PLWH groups compared to the UC group, revealing a deep perturbation of T cell homeostasis in the context of HIV infection. Moreover, the heatmaps of fold-change of expression of each sample over the median expression in the UC group showed a clear separation between each PLWH group and the UC group (Appendix A).

### 2.3. Uncontrolled Viral Replication Is Associated with a Deep Perturbation of CD4 and CD8 T Cell Homeostasis

The highest perturbation of T cell homeostasis was observed in PLWH with uncontrolled viral replication (offART group). Regarding CD4 T cells, 18 different clusters were differentially expressed: most of them (14) decreased and only 4 increased with respect to UCs (Figure 1). Among the decreased clusters, four (C00, C02, C12, C44) were formed by naïve cells (CD45RA^+^CCR7^+^CD28^+^) expressing CD127 and CD38 markers, with two of them also expressing CD31 (RTE cells). Another seven clusters (C09, C13, C17, C22, C24, C26, C38) were formed by central memory (CM) cells (CD45RA^−^CCR7^+^CD28^+^): three with a Th1 phenotype (CXCR3^+^CCR6^−^), three with a Th2 phenotype (CXCR3^−^CCR6^−^), and another with a mixed Th1/Th17 phenotype (CXCR3^+^CCR6^+^). All seven clusters of central memory phenotypes expressed low levels of activation and exhaustion. Another two decreased clusters (C36 and C39) were formed by pTfh cells (Appendix A). Among the increased CD4 clusters, one was formed by central memory cells with high expression of activation and exhaustion (C28) and another by TEMRA (CD45RA^+^CCR7^−^CD28^−^) cells with high expression of activation, apoptosis, senescence, and exhaustion (C34) (Appendix A).

Regarding CD8 T cells, 15 different clusters were differentially expressed: 10 decreased and 5 increased compared to the UC group (Figure 2). Among the decreased clusters, six (C00, C01, C12, C26, C28, C45) were formed by naïve cells expressing CD127, with some of them also expressing Tim3 or CD57. Another three clusters (C22, C23, C39) were also formed by naïve cells co-expressing CD127 and CD95 markers. Among the increased clusters, two of them (C06, C32) were formed by effector memory (EM) (CD45RA^−^CCR7^−^CD28^−^) cells with high expression of activation and exhaustion markers. The other three clusters (C25, C30, C35) were formed by effector memory re-expressing CD45RA (TEMRA) cells with high expression of activation and senescence markers.

As another approach to analyze the distribution of CD4 and CD8 T cell clusters between the PLWH groups and the UC group as a reference, a random forest and hierarchical clustering were carried out using the levels of the different clusters (Appendix A). The random forest model showed very good discrimination between the offART and UC groups. Moreover, a heatmap of the hierarchical clustering showed a clear difference in the profiles of CD4 and CD8 T cell cluster expression levels when comparing the offART and UC groups (Appendix A).

### 2.4. The Majority of T Cell Homeostasis Disturbances Are Restored with Long-Term ART

PLWH with ART-mediated long-term suppression of viral replication (onART group) presented the lowest perturbation of T cell homeostasis, with a profile of CD4 and CD8 T cells approaching the profile of the UC group. However, some perturbation still remained, with one cluster of CD4 (C45) (Figure 1) and two clusters of CD8 (C27 and C30) (Figure 2) T cells differentially expressed in the onART compared to the UC group. Of note, two of the clusters (C30 and C45) were formed by TEMRA cells expressing high levels of activation, exhaustion, and senescence (Appendix A).

### 2.5. PLWH with Spontaneous Control of HIV Replication Show a Deep Perturbation of CD4 and CD8 T Cell Clusters

The most interesting finding came from the LTEC group, in which a large number of CD4 and CD8 T cell clusters were differentially expressed compared to the UC group, even though plasma HIV load was undetectable, as in the onART group.

Regarding CD4 T cells, fifteen different clusters were differentially expressed in LTECs: eight decreased and seven increased (Figure 1). The decreased clusters included three clusters formed by Th2 naïve cells expressing CD127 and CD38 (C02, C18, and C12); two clusters formed by CM cells expressing CD95 and CD25 (C24 and C26); two clusters formed by pTfh (CXCR5+) cells (C36 and C39); and, lastly, one cluster formed by Treg (CD127-CD25+) cells (C43). The increased clusters included two clusters formed by Th2 naïve cells expressing CD127 and CD38 (C00 and C44); four clusters formed by CM cells with variable levels of activation and exhaustion (C13, C21, C38, and C42), three of them with a Th1 phenotype; and one cluster formed by TEMRA cells with high levels of activation, exhaustion, and senescence (C51) (Appendix A). It is important to clarify that although LTECs exhibited both decreased (C02, C18, C12) and increased (C00, C44) clusters of Th2 naïve CD4 T cells, there were some important phenotypic differences between them. Thus, among the decreased clusters, two of them (C18, C12) expressed moderate levels of the activation marker CD25 and the other decreased cluster (C02) expressed moderate levels of the exhaustion marker CTLA4. In contrast, the two increased clusters (C00, C44) lacked the expression of CD25 or CTLA4. There were also both decreased (C24, C26) and increased (C13, C21, C38, C42) clusters of CM CD4 T cells in LTECs, but, again, there were some important phenotypic differences between the decreased and the increased central memory clusters, mainly in the expression profiles of activation and exhaustion markers (Appendix A).

Regarding CD8 T cells, as many as thirteen clusters were differentially expressed in the LTEC group: seven increased and six decreased (Figure 2). Increased clusters were mainly formed by cells with TEMRA (C11, C30, C38) or EM (C06) phenotypes with high expression of activation and exhaustion markers. Among the decreased clusters, most of them were formed by cells with naïve phenotypes expressing CD127, with three of them (C00, C01, C04) also expressing Tim3 and two of them (C22, C23) also expressing CD95 (Appendix A). As for the offART group, the random forest model showed very good discrimination between the LTEC and UC groups, and a heatmap of the hierarchical clustering showed a clear difference in the profiles of CD4 and CD8 T cell cluster expression levels when comparing the LTEC and UC groups (Appendix A).

Notably, the profile of differentially expressed clusters (with respect to the UC group) partially overlapped between LTEC and offART groups (Appendix A). Of the 15 CD4 T cell clusters altered in LTECs, 10 (67%) were in common with the clusters altered in the offART group. However, only six of these common clusters (C02, C12, C24, C26, C36, C39) were altered in the same direction in the LTEC and offART groups. All six clusters were decreased in both groups and consisted of naïve (C02, C12), CM (C24, C26), and pTfh (C36, C39) cells. In contrast, the other four clusters (C00, C13, C38, C44) were altered in opposite directions in the LTEC and offART groups (increased in LTECs and decreased in offART). As for CD8 T cell clusters, of the 13 clusters altered in LTECs, 7 (54%) were in common with the clusters altered in the offART group, and all of them altered in the same direction in the LTEC and offART groups. Of these common clusters, decreased clusters were formed by naïve cells (C00, C01, C22, C23), whereas increased clusters were formed by senescent and/or exhausted TEMRA cells (C30) or by EM cells (C06).

### 2.6. A Specific Profile of CD4 and CD8 T Cell Clusters Associated with the Spontaneous Control of HIV Replication

Although the profiles of T cell homeostasis alteration presented some similarities in the LTEC and offART groups, different findings demonstrated the existence of a specific profile of T cell homeostasis alteration associated with the LTEC group. First, there were some CD4 and CD8 T cell clusters that were significantly altered (with respect to the UC group) only in the LTEC group (Appendix A). Five CD4 T cell clusters were significantly altered only in LTECs (C18, C21, C42, C43, C51), although C18 and C21 were altered in the same direction also in the offART group, without reaching statistical significance. Thus, only C42, C43, and C51 were distinctive for the LTEC group. Interestingly, C43 (decreased in LTECs) was formed by Treg cells with a mixed Th1/Th17 phenotype, whereas C42 (increased in LTECs) was formed by CM cells with a Th1 phenotype expressing high levels of activation (Appendix A). As for CD8 T cell clusters, six clusters presented a significant alteration only in LTECs (C04, C11, C15, C26, C29, C38), although three of them (C04, C11, C38) were also altered in the same direction in the offART group, without statistical significance. Thus, only clusters C15, C26, and C29 were distinctive for the LTEC group. C15 (decreased in LTECs) was formed by senescent TEMRA cells, whereas C29 (increased in LTECs) was formed by cells with a mixed Tc1/Tc17 phenotype (CXCR3+CCR6+) expressing CD127, and C26 was formed by naïve cells with a Tc1 phenotype (Appendix A).

Secondly, there were many T cell clusters differentially expressed between the LTEC and offART groups (Appendix A), including twelve clusters of CD4 T cells and nine clusters of CD8 T cells. Among the CD4 T cell clusters, increased clusters in LTECs were formed by clusters of naïve cells (C00, C44), CM/Th1 cells (C14, C17, C38), CM/Th2 cells (C13), and CM/Th1-Th17 cells (C09, C42). Decreased clusters in LTECs were formed by clusters of naïve cells (C08, C37), one of them with high levels of exhaustion (C37); exhausted CM/Th1cells (C28); and senescent TEMRA cells (C34) (Appendix A). Regarding CD8 T cell clusters, increased clusters in LTECs were formed by clusters of naïve cells (C01, C12, C26, C28) and cells with a mixed maturation stage expressing CD127 (C20, C29, C39). Decreased clusters in LTECs were formed by clusters of senescent/activated TEMRA cells (C30) and activated cells with a mixed maturation stage (C32) (Appendix A).

Thirdly, several clusters were differentially expressed between the LTEC and onART groups, despite undetectable plasma HIV viremia in both groups (Appendix A). With regard to CD4 clusters, eight clusters were down and four clusters were up for LTECs. The decreased clusters included naive (C02, C18, C37, C43), CM (C26), TEMRA (C35), and pTfh (C36, C39) clusters, while the increased clusters included naive (C00, C44) and CM (C13, C42) clusters. As for the CD8 clusters, LTECs showed a decrease in several naive clusters (C00, C04, C22, C23) and one TEMRA cluster (C15) and an increase in one naive cluster (C26), one EM cluster (C06), and two clusters composed of cells with a mixed maturation stage (C29, C32) (Appendix A).

Lastly, there were a few T cell clusters differentially expressed in the LTEC group when compared with all the other study groups (UC, offART, and onART groups). Among CD4 cells, clusters C00 and C44 (formed by naïve cells) were greatly increased in LTECs compared to the UC, offART, and onART groups. Clusters C13, C38, and C42 (formed by CM cells) were also greatly increased in LTECs compared to the rest of the groups (Figure 1). Regarding CD8 cells, cluster C26 (formed by naïve cells) was greatly increased and cluster C15 (formed by senescent TEMRA cells) was decreased in LTECs compared to the rest of the groups (Figure 2).

### 2.7. Correlations between T Cell Clusters and Inmmuno-Virological Parameters

The potential associations of levels of CD4 and CD8 T cell clusters with virological (pVL) or immunological (CD4 counts) parameters were analyzed using non-parametric tests (Spearman rho correlation coefficient). In the offART group, the level of pVL was positively correlated with two clusters of naïve CD4 T cells (C80, C19) and two clusters of naïve CD8 T cells (C16, C26), whereas negative correlations were observed with CM clusters of CD4 T cells (C14, C21), pTfh clusters of CD4 T cells (C10, C36), and TEMRA clusters of CD8 T cells (C09, C19) (Figure 3). Regarding correlations with CD4 counts, the profile of correlations was different for the different PLWH groups (Figure 4). The offART and onART groups presented several positive correlations with naïve clusters of CD4 T cells; additionally, the offART group presented a negative correlation with pTfh and Treg clusters of CD4 T cells. In contrast, the LTEC group presented positive correlations between CD4 counts and several clusters of CD8 T cells, mainly of EM or TEMRA phenotypes, something that was also observed to a lesser extent in the offART group (Figure 4).

## 3. Discussion

In order to identify the phenotypic traits of T cells linked to the long-term spontaneous control of viral replication, this study is the first, to our knowledge, to perform a thorough phenotypic analysis of T cell homeostasis in PLWH who are long-term elite controllers (LTECs) compared with that found in non-controller PLWH with uncontrolled viremia and in non-controller PLWH with treatment-suppressed viremia. Our study’s main conclusions are as follows: (a) the thorough phenotypic and multidimensional analysis approach allowed us to identify distinct variations in T cell homeostasis profiles between the various PLWH groups; (b) T cell homeostasis disturbances are most pronounced in PLWH with uncontrolled viral replication; (c) the majority of T cell homeostasis disturbances are restored by long-term ART-mediated suppression of viral replication; (d) interestingly, T cell homeostasis is significantly altered in LTECs compared to PLWH with treatment-suppressed viremia; (e) the T cell homeostasis profile that was specifically associated with the LTEC group was characterized by a decrease in Treg cell clusters; a decrease in cell clusters expressing high levels of activation, exhaustion, and senescence; and an increase in clusters of naïve cells and central memory cells.

Uncontrolled viral replication was linked to a significant disruption in the phenotypic profiles of T cells. In the offART group, alterations were found in 18 out of 57 (32%) CD4 T cell clusters and in 15 out of 54 (28%) CD8 T cell clusters when compared to the UC reference group. The most significant alterations in the offART group were characterized by decreases in several CD4 T cell clusters of naïve phenotype and of CM phenotype expressing low levels of activation and exhaustion as well as decreases in clusters of pTfh cells; there were also decreases in naïve clusters of CD8 T cells and increases in activated and exhausted EM and TEMRA clusters of CD8 T cells. Altogether, these results point to a significant impairment of T cell homeostasis in PLWH with uncontrolled HIV replication, in agreement with previous studies [15,16,17]. There were several correlations between T cell clusters and levels of HIV viremia, suggesting that many of these alterations may be the consequence of viral replication. Thus, levels of CM and pTfh CD4 T cells were negatively correlated with plasma viremia, in agreement with the diminished levels of these clusters in the offART group. It is interesting that, in spite of the decrease in several naïve clusters of CD4 T cells, some naïve clusters showed a positive correlation with HIV viremia (clusters C08 and C19), which supports the functional diversity of clusters with naïve phenotype. This was also observed for two naïve clusters of CD8 T cells (C16 and C26), which correlated positively with HIV viremia. Of note, the naïve clusters showing a positive correlation with HIV viremia also expressed CD31 (a marker of recent thymic emigrants) and lacked expression of the rest of the markers analyzed (activation, exhaustion, senescence), suggesting that these clusters may represent the thymus´s attempt to maintain the pool of naïve T cells [18,19]. Of note, some clusters of naïve CD4 T cells (C01 and C19) were positively correlated with CD4 counts, which supports a relevant role of these clusters in the maintenance of the CD4 T cell pool. A similar phenomenon was observed for TEMRA clusters of CD8 T cells, with some of them (C09 and C19) showing a negative correlation with HIV viremia, even though there was an increase in several TEMRA clusters in the offART group, which supports the functional heterogeneity of CD8 T cell clusters with TEMRA phenotype.

The majority of T cell homeostasis alterations seen in PLWH with uncontrolled viremia were reversed in PLWH with ART-mediated control of HIV replication, suggesting that these alterations may be the result of high viral replication. Interestingly, the few clusters that remained altered in the onART group were TEMRA clusters expressing high levels of activation, exhaustion, and senescence, which suggests the persistence of low viral replication driving the expansion of these highly differentiated T cell clusters [20,21]. Alternatively, the expansion of these highly differentiated subsets of T cells in the onART group may have been driven by the presence of cytomegalovirus (HCMV) infection [22]. We did not analyze either the serostatus of HCMV or the presence of HCMV reactivation and, thus, we could not test this hypothesis.

The most interesting findings of our study were those found in the LTEC group. In contrast to the onART group (in which the suppression of viremia was mediated by ART), the spontaneous control of HIV replication was associated with a deep perturbation of T cell homeostasis. Overall, our findings agree with those of previous studies addressing specific aspects of T cell homeostasis in different cohorts of PLWH who are ECs, such as the distribution of the maturation stage of T cells [23,24,25]; the activation, exhaustion, and senescence of T cells [26,27,28]; and Treg cell levels [29,30]. Our study confirms and extends these previous results by the analysis of a well-characterized cohort of elite controllers with long-term control of HIV infection, in which a comprehensive phenotypic analysis of T cells and a dimensionality reduction approach allowed us to perform an in-depth analysis of the homeostasis of this cell population. Moreover, this approach allowed us to find a specific profile of T cell homeostasis perturbation associated with the spontaneous control of HIV infection. Several findings from our study point to a profile of T cell homeostasis alteration specific to LTECs: (a) changes in different clusters of T cells that were only found in the LTEC group (when compared to the UC group as a reference); (b) differences in T cell cluster levels between the LTEC and onART groups, despite undetectable plasma viremia in both groups; (c) changes in several clusters of T cells that differentiated the LTEC group from the rest of the groups.

Of note, many of the perturbations observed in the LTEC group were similar to those observed in the context of uncontrolled viral replication. These perturbations could be driven by the existence of residual viral replication in LTECs at higher levels than in the ART-mediated control of HIV replication, a fact that has been previously reported [31,32,33]. In agreement with this hypothesis, previous studies have shown that antiretroviral treatment in elite controllers decreases T cell activation and exhaustion [34,35]. Alternatively, the existence of T cell homeostasis perturbations in LTECs may be a consequence of the existence of a potent and effective HIV-specific immune response. Interestingly, LTECs showed increased levels of non-senescent TEMRA clusters of CD8 T cells expressing low levels of activation and exhaustion (C11), something that was not observed in the offART group, in which the increased clusters of TEMRA CD8 T cells expressed high levels of activation, exhaustion, and senescence (C25, C30, C35). Taking into account the more preserved functionality of cluster C11 (non-senescent and non-exhausted), it is tempting to speculate that the expansion of this cluster in LTECs may be associated with the better anti-HIV CD8 T cell response reported in PLWH able to spontaneously suppress HIV replication [3]. Another perturbation of T cell homeostasis in the LTEC group that could be associated with the better HIV-specific immune response in this group is the existence of several expanded clusters of CM CD4 T cells in LTECs when compared not only to the UC group but also to the offART and onART PLWH groups. CM CD4 T cells are necessary to maintain a functional anti-viral CD8 T cell response [36]. Moreover, all these CM clusters expressed CD127, a surrogate marker of preserved homeostatic potential [37]. Lastly, some characteristics of T cell homeostasis specific of the LTEC group could be associated with the ability of this group of PLWH to maintain control of immunological progression (depletion of the CD4 T cell pool), among them the expansion of clusters of naïve CD4 (clusters C00, C44) and naïve CD8 (C26) T cells). C00 is a cluster of naïve cells expressing CD31 (recent thymic emigrants) and CD127 (proliferative potential), suggesting a high capacity of this cluster to repopulate the CD4 T cell pool. Cluster C44 has the same phenotype as C00, except that it lacks the expression of CD31, suggesting that this cluster may originate from the peripheral expansion of CD31+ naïve cells. Of note, a previous study of a cohort of elite controllers reported that a low level of naïve CD4 T cells is associated with HIV disease immunological progression [14].

In summary, the results of our study demonstrate the existence of a specific profile of T cell homeostasis in the population of PLWH that spontaneously control HIV disease in the long term (LTECs). This specific profile may be the consequence of different factors operating in this special population of PLWH: (a) the existence of low-level HIV replication; (b) the existence of a potent and efficient virus-specific T cell response; (c) the ability to halt disease progression by maintaining an adequate pool of CD4 T cells.

## 4. Materials and Methods

### 4.1. Study Design and Participants

This was a cross-sectional study including adult participants with chronic HIV infection (people living with HIV, PLWH) and HIV-seronegative healthy volunteers (uninfected controls, UC group). Three different groups of PLWH participants were included: PLWH with long-term spontaneous control of HIV infection (long-term elite controllers, LTEC group); PLWH non-controllers on antiretroviral treatment (ART) with an undetectable plasma HIV viral load (pVL) for at least five years at the time of inclusion in the study (onART group); and PLWH non-controllers off ART with detectable HIV pVL (offART group). The LTEC group was selected from the Spanish ECRIS database, a multicenter cohort of HIV elite controllers (ECs) launched in 2013 and described elsewhere [10]. The ECsRIS cohort includes patients from the long-term non-progressors (LTNPs) cohort, the AIDS research cohort (CoRIS), and several clinical centers in Spain (Appendix A). From this ECsRIS cohort, LTECs were defined as those ECs maintaining control (both virological and immunological) for a minimum of five years and throughout their whole follow-up period. Virological control was defined as having more than 90% of HIV pVL during the follow-up below the limit of detection. Immunological control was defined as having stable CD4 counts during follow-up. All study participants signed written informed consent. This study was approved by the Ethical Review Board of the Instituto de Investigación Sanitaria-Fundación Jiménez Díaz, Madrid, Spain (Approval ID: PIC097-19_FJD) and carried out in accordance with the Declaration of Helsinki.

### 4.2. Cell Samples

Blood samples were collected by venipuncture in tubes containing ethylenediaminetetraacetic acid (EDTA) as an anticoagulant, sent on the same day to the Spanish HIV HGM BioBank (http://hivhgmbiobank.com/?lang=en), immediately processed to obtain peripheral blood mononuclear cells (PBMCs) by gradient centrifugation over a ficoll-paque solution (Sigma-Aldrich, St. Louis, MA, USA), and kept frozen in liquid nitrogen. PBMC samples were kindly provided by the Spanish HIV HGM Biobank.

### 4.3. Multiparameter Flow Cytometry

One million PBMCs were stained with a panel of twenty-one different monoclonal antibodies (plus a viability dye) (Appendix A). A detailed staining protocol is given in the Appendix A. After staining, cells were acquired using an Aurora Spectral flow cytometer (Cytek Biosciences, Fremont, CA, USA), and a minimum of 200,000 events were acquired for further analysis. Data analysis was performed using FCS Express 7 v.7 (DeNovo software, Pasadena, CA, USA). Dead cells were excluded using the live/dead viability dye, and live lymphocytes were gated based on forward (FSC) and side (SSC) scatter. Lastly, single live lymphocytes were selected using FSC width vs. FSC height. Starting from the population of single live lymphocytes, a gate was placed to select T cells (CD3^+^ lymphocytes) and then CD3^+^CD4^+^CD8^−^ or CD3^+^CD8^+^CD4^−^ T cells were selected for further analysis.

### 4.4. Preparation of Flow Cytometry Data for Unsupervised Multidimensional Analysis

Flow cytometry data from each study participant were manually gated using FCS Express 7 software as described above. Using the CD3^+^CD4^+^CD8^−^ and CD3^+^CD8^+^CD4^−^ gates, each fcs. file was split into two different files: one containing only the CD3^+^CD4^+^CD8^−^ events and another containing only the CD3^+^CD8^+^CD4^−^ events. Next, these new fcs. files were annotated using FCS Express 7 to include information about the positivity/negativity of each event contained in the file for each of the 18 different cell surface markers (lineage markers CD3, CD4, and CD8 were excluded from the annotation). These fcs. files were used for dimensionality reduction and clustering analysis.

### 4.5. Unsupervised Analysis of Flow Cytometry Data

All pre-gated data files containing CD3^+^CD4^+^CD8^−^ and CD3^+^CD8^+^CD4^−^ cells were included for unsupervised multidimensional analysis [38], including batch correction, dimensionality reduction, and clustering analysis. R software (version 4.1.1) was used for the implementation of the automated pipeline. The workflow for this analysis is shown in Appendix A.

First, batch effects were corrected using the CytoNorm algorithm [39]. By taking advantage of the existence of the same control sample acquired in each batch or experiment along with the rest of the samples, non-biological variability was estimated and removed from the data before their unbiased analysis. Each file was subsampled to 25,000 cells to facilitate the proper processing of all samples at once and balance the contribution of each sample to the total number of cells analyzed. After transforming the data with the arcsinh function (cofactor = 6000), marker expression levels were jointly scaled via Zscore normalization. The whole panel of fluorescent markers (except the lineage markers CD3, CD4, and CD8) were used in the downstream analysis using the Seurat R package [40]. For the detection of cell subpopulations sharing a similar expression fingerprint, we used Louvain’s algorithm [41], which identifies clusters by modularity optimization from a shared nearest neighbor graph (SNN). A level of cluster granularity corresponding to a resolution parameter value of 2.0 was used. To visualize single-cell measurements and clustering results, we used 2-dimension representations obtained with nonlinear dimensionality reduction techniques. T-distributed stochastic neighbor embedding (tSNE) maps were obtained using the FFT-accelerated Interpolation-based t-SNE (FIt-SNE) algorithm [42] (parameters: perplexity = 30, theta = 0.5, distance = Euclidean, rest default parameters). Cell frequencies in each cluster were computed and compared between each pair of groups using the Wilcoxon test. *p*-values were corrected using the Holm method. We also computed fold-change with respect to a reference group (typically the UC group) to represent the relative changes in cluster proportions between groups.

### 4.6. Statistical Analysis

The clinical and epidemiological characteristics of the different study groups were expressed as median and interquartile ranges, and differences were tested using non-parametric tests (Kruskall–Wallis test). Levels of the different clusters of CD4 and CD8 T cells were compared among the different study groups, as explained above. Moreover, using the levels (proportion) of the different CD4 and CD8 T cell clusters in each study volunteer, a random forest and hierarchical clustering analysis were performed as classification methods in order to ascertain the T cell clusters involved in discriminating between pairs of study groups. The Boruta algorithm was applied to select the most important variables for classification. Variables selected by Boruta were used to carry out the random forest and hierarchical clustering. The number of trees and variables for the random forest were optimized by minimizing the out-of-bag error. An internal validation, using the leave-one-out method, was conducted to assess the classification capabilities of the random forest. Hierarchical clustering was performed using the Euclidean distance and Ward’s agglomerative method and graphically represented by a heat map with dendrograms.

## Figures and Tables

**Figure 1 ijms-25-05937-f001:**
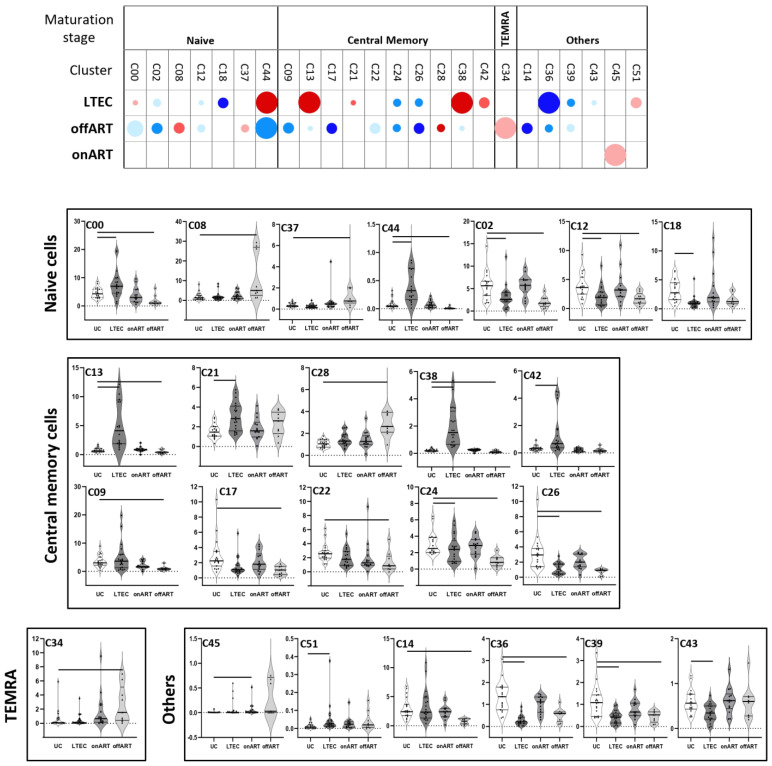
The upper graph of the figure shows a schematic representation (bubble diagram) of CD4 T cell clusters, with significant differences between the PLWH groups and the UC group. Clusters are grouped according to maturation stage (defined by CD45RA and CCR7 markers). Each bubble in the diagram represents a significant difference with respect to the UC group. The size of the bubble (from the smallest to the biggest size) indicates the degree of difference in fold-change: <2, 2–3, 3–4, 4–5, and >5. Red colors indicate an increase and blue colors indicate a decrease with respect to the UC group. The level of statistical significance (corrected *p*-value) is indicated by the color tone: 0.05–0.01, 0.01–0.001, and <0.001 for the light, medium, and dark tones, respectively. The lower graphs of the figure show violin plot graphs of the levels (expressed as a percentage of total CD4 T cells) of each cluster in the four study groups. As in the upper graph, clusters are grouped according to maturation stage. In each violin plot graph, bars inside the plots indicate those PLWH groups that showed a statistically significant difference with respect to the UC group.

**Figure 2 ijms-25-05937-f002:**
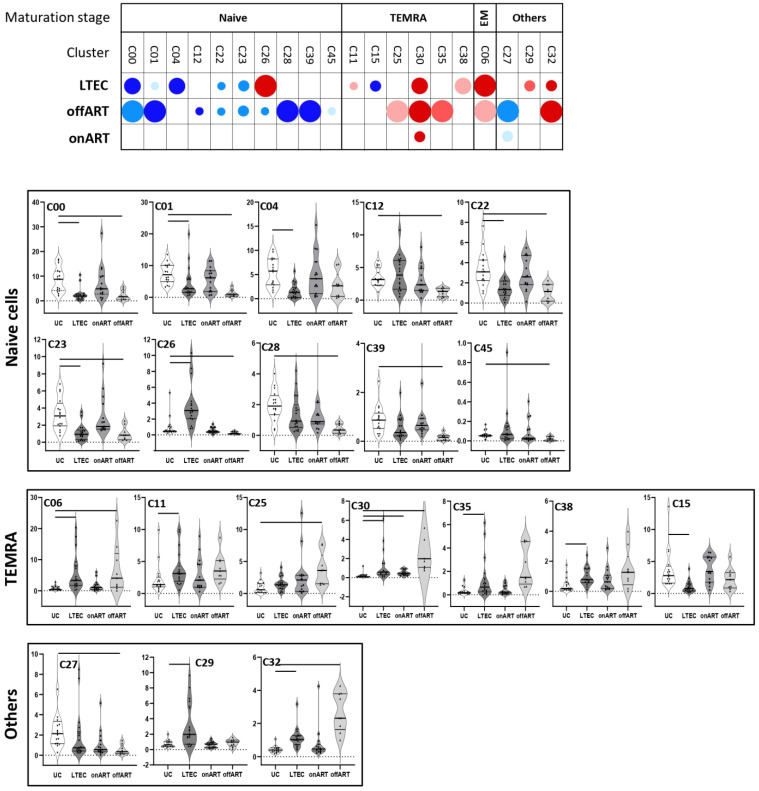
The upper graph of the figure shows a schematic representation (bubble diagram) of CD8 T cell clusters, with significant differences between the PLWH groups and the UC group. Clusters are grouped according to maturation stage (defined by CD45RA and CCR7 markers). Each bubble in the diagram represents a significant difference with respect to the UC group. The size of the bubble (from the smallest to the biggest size) indicates the degree of difference in fold-change: <2, 2–3, 3–4, 4–5, and >5. Red colors indicate an increase and blue colors indicate a decrease with respect to the UC group. The level of statistical significance (corrected *p*-value) is indicated by the color tone: 0.05–0.01, 0.01–0.001, and <0.001 for the light, medium, and dark tones, respectively. The lower graphs of the figure show violin plot graphs of the levels (expressed as a percentage of total CD8 T cells) of each cluster in the four study groups. As in the upper graph, clusters are grouped according to maturation stage. In each violin plot graph, bars inside the plots indicate those PLWH groups that showed a statistically significant difference with respect to the UC group.

**Figure 3 ijms-25-05937-f003:**
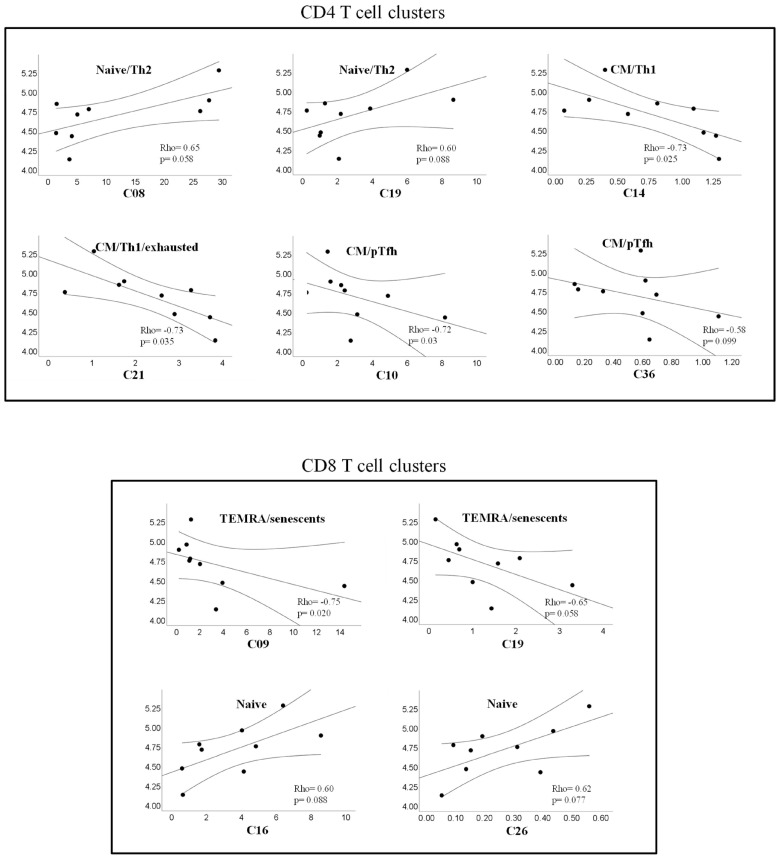
Scatter plots showing correlations between HIV plasma viral load and levels of different clusters of CD4 (upper plots) or CD8 (lower plots) T cells. Correlations with plasma HIV load were analyzed in the offART group. In all plots, the *x*-axis represents the level of each cluster expressed as a percentage of total CD4 or CD8 T cells and the *y*-axis represents the HIV plasma viral load expressed as copies/mL (log scale). The Spearman’s Rho correlation coefficient and *p*-value are given inside each scatter plot and the legend over each plot describes the phenotype of cells forming that cluster.

**Figure 4 ijms-25-05937-f004:**
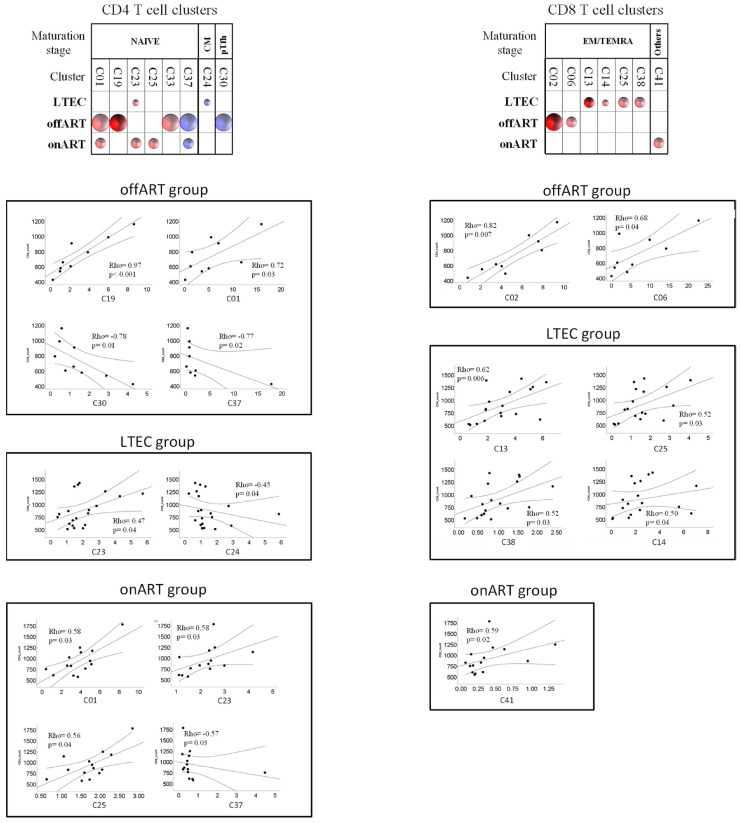
Schematic representation (bubble diagram) of bivariate correlations (Spearman rho coefficient) between CD4 (upper left diagram) or CD8 (upper right diagram) T cell clusters and CD4 counts in the different PLWH groups. Each bubble in the diagram represents a positive (red color) or negative (blue color) correlation. The size of the bubble (from the smallest to the biggest size) indicates the value of the Spearman rho coefficient: <0.5, 0.5–0.7, and >0.7. The level of statistical significance (*p*-value) is indicated by the color tone: 0.1–0.05, <0.05–0.01, and <0.01 for the light, medium, and dark tones, respectively. T cell clusters are grouped according to maturation stage. Below the diagrams, scatter plots of correlations between CD4 counts (*y*-axis on the plots) and the levels of different clusters of CD4 (left) and CD8 (right) T cells are shown. The Spesarman’s rho coefficient and *p*-value are given inside each scatter plot.

**Table 1 ijms-25-05937-t001:** Characteristics of groups included in this study.

Characteristic	LTEC (n = 22)	onART (n = 15)	offART (n = 10)	UC (n = 20)	*p*-Value
Age (years)	44 [35–49]	44 [42–49]	43 [36–49]	43 [36–49]	0.266
Sex (% of males)	59 *	80	100	50 *	**0.025**
Years since HIV diagnosis	15 * [6–20]	12 [5–15]	5 [3–9]	**NA**	**0.027**
Years as EC	13 [7–16]	NA	NA	NA	NA
Years on ART	NA	6 [3–9]	NA	NA	NA
Plasma HIV load (copies/mL)	50	50	57,937 [28,817–80,676]	NA	NA
CD4 count (cells/μL)	837 [603–1210]	820 [599–1127]	625 [518–920]	NA	0.200

Data are given as median [Q1–Q3], except sex that is expressed as a percentage. Statistical differences between the four study groups were tested by Kruskal-Wallis test for continuous variables and by Chi-square test for sex. Statistical significance was considered for *p*-values below 0.05 (in bold in the table). NA: not appicable. An asterisk denotes statistically significant differences (*p* < 0.05) with respect to the offART group.

## Data Availability

The raw data supporting the conclusions of this article will be made available by the authors upon request.

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
