# Peer review of "T Cell Homeostasis Disturbances in a Cohort of Long-Term Elite Controllers of HIV Infection"

_ijms, 2024, doi:10.3390/ijms25115937_

Round 1
Reviewer 1 Report
Comments and Suggestions for Authors
The authors present a well-planned, methodologically supported work, with a large number of results. The writing in general seems adequate to me, although the English requires some improvement, it does not affect general understanding. However, given the large number of results, the authors should reconsider whether some of the files in the supplementary material would be better placed in the main text, especially the correlations that the authors present.
However, some important suggestions are the following:
1. Table 1: Authors should include a post hoc analysis to determine the direction of significance.
2. The resolution of all figures must be profoundly improved, as it greatly affects the correct visualization of the data. Without this, the quality of the results cannot be adequately evaluated.
3. However, in figures (1 and 2), the violin plots do not show significance bars, which the authors indicate with colors. For clarity, authors should add significance bars, not colors.
Author Response
Comment 1: …However, given the large number of results, the authors should reconsider whether some of the files in the supplementary material would be better placed in the main text, especially the correlations that the authors present
Response: we agree with the reviewer and accordingly we have included two supplementary files of the old version of manuscript (supplementary file 10 and supplementary file 11) as new files in the main text (figure 3 and figure 4)
Comment 2: Table 1: Authors should include a post hoc analysis to determine the direction of significance.
Response: we agree with the reviewer and accordingly we have included this new information in the revised version of table 1. This revised version of the manuscript incorporates this new revised version of table 1.
Comment 3: The resolution of all figures must be profoundly improved, as it greatly affects the correct visualization of the data. Without this, the quality of the results cannot be adequately evaluated.
Response: we agree with the reviewer that figures appearing in the old version of the manuscript are very difficult to read due to the very low resolution they show. Following reviewer suggestion, we have changed the format of figures appearing in the main text (figure 1, figure 2, figure 3, and figure 4) to increase their resolution and make them readable.
Comment 4: However, in figures (1 and 2), the violin plots do not show significance bars, which the authors indicate with colors. For clarity, authors should add significance bars, not colors.
Response: following the reviewer suggestion, we have deleted blue and red colors and have used significance bars in the violin-plots of figures 1 and 2. These new revised figures are included in the revised version of the manuscript.
Reviewer 2 Report
Comments and Suggestions for Authors
Summary of the main findings of the study:
The study investigated perturbations in T cell subsets among individuals comprising 47 HIV patients and 20 uninfected controls. The HIV cohort included long-term elite controllers (LTEC, n=22), non-controllers on antiretrovirals (onART, n=15), and non-controllers not receiving antiretrovirals (offART, n=10), compared to uninfected controls (UC, n=20). The offART group exhibited the highest number of perturbations, with 18 CD4 and 15 CD8 clusters showing significant differences, whereas these changes were absent in the onART group. Within the LTEC group, increases were observed in clusters of naïve T-cells, non-senescent effector CD8 cells, and central memory CD4 cells.
Comments:
· Considering the complexity of the markers involved, it is advisable to create a table elucidating the usage and interpretation of marker combinations. For instance, clarifying how CD28 is employed in various combinations to discern maturation, senescence, and apoptosis would enhance comprehension. The table should also outline the expected expression of markers for specific T cell subsets.
· Supplementary Table 1: Address formatting issues, such as row height, which may obscure row content.
· Regarding lines 192-197: Clarify the interpretation of findings concerning differentially expressed clusters observed to be decreased in LTEC versus UC, notably Th2 na√Øve cells and central memory, which exhibit both decreased and increased levels simultaneously using the same markers.
Author Response
Comment 1: Considering the complexity of the markers involved, it is advisable to create a table elucidating the usage and interpretation of marker combinations. For instance, clarifying how CD28 is employed in various combinations to discern maturation, senescence, and apoptosis would enhance comprehension. The table should also outline the expected expression of markers for specific T cell subsets.
Response: we agree with the reviewer and accordingly we have included a new supplementary table (supplementary table 1) showing the information regarding the phenotypic definition (markers combination) of different functional traits of T cells as well as some references supporting the definition. This new table is cited in section 2.2 of Results (“Unsupervised Multidimensional Analysis of Flow Cytometry Data Yielded Several CD4 and CD8 T Cells Clusters that Differentiate the Study Groups”). The expected expression of these specific subsets is not included in the table, given that information on this issue for many of the subsets included in the table is lacking. Nonetheless, since our study includes a group of healthy uninfected controls, the values obtained in our study could be taken as reference values for these specific subsets. However, since data analysis was carried out using dimensionality reduction and clustering (as explained in the methods section of the manuscript), data regarding the levels of the specific subsets of T cells appearing in this new supplementary table, is lacking.
Comment 2: Supplementary Table 1: Address formatting issues, such as row height, which may obscure row content.
Response: we agree with the reviewer and accordingly we have expanded the row height of the table so that the information can be readable. This new revised version of supplementary table 1 (now renamed supplementary table 2) is included in the supplementary material of the revised version of the manuscript.
Comment 3: Regarding lines 192-197: Clarify the interpretation of findings concerning differentially expressed clusters observed to be decreased in LTEC versus UC, notably Th2 naïve cells and central memory, which exhibit both decreased and increased levels simultaneously using the same markers.
Response: we agree with the reviewer that it is confusing to say that Th2 naïve CD4 clusters are at the same time increased and decreased in LTEC group. This is also the case for some central memory clusters of CD4 cells. However, although similar in many aspects, the increased and decreased clusters (either Th2 naïve clusters or central memory clusters) are different in come other phenotypic aspects. This is clarified in the revised version of the manuscript (in section 2.5 of Results “PLWH with Spontaneous Control of HIV Replication Show a Deep Perturbation of CD4 and CD8 T Cells Clusters”).